# Design and Analysis of Novel Linear Oscillating Loading System

**Zongxia Jiao** [1,2,3,*], **Yuan Cao** [2], **Liang Yan** [1,2,3,*], **Xinglu Li** [2] and **Lu Zhang** [2]

1   Science and Technology on Aircraft Control Laboratory Beihang University, Beijing 100191, China
2   School of Automation Science and Electrical Engineering, Beihang University, Beijing 100191, China
3   Ningbo Institute of Technology, Beihang University, Beijing 100191, China
*   Correspondence: zxjiao@buaa.edu.cn (Z.J.); lyan1991@gmail.com (L.Y.); Tel.: +86-181-0334-3996 (Z.J.)



**Featured Application:** **The paper gives design and analysis of novel linear oscillating loading system to illustrate dynamic performance of linear oscillating motor with different types of external load force. The linear oscillating loading system shows the linear oscillating motor can operate at large force load, which will have a prosing improvement on the linear oscillating motor application to linear pump or other equipment that is driven directly.**

**Abstract:** Although linear motor has vital and potential applications in air compressors, hydraulic pumps, earphones and electric vehicles because of its good reliability, high power density and convenient maintenance, most researchers rarely concentrate on the dynamic performance of the linear oscillating motor with external force loads. It is essential to study the dynamic performance of the linear oscillating motor with accurate and multi-mode force loads. In this paper, a novel linear oscillating loading system is proposed and the loading system structure is depicted. Then, a mathematical model is built to match the simulation analyses of the dynamic performance of the linear oscillating motor with multi-mode external force loads. Moreover, the linear oscillating loading system platform is built and experiments are undertaken to verify the simulation analyses about the dynamic performance and efficiency with respect to different external force loads, and the simulation and experimental results show good agreement and will have promising significance for linear oscillating motor research and applications.

**Keywords:** linear oscillating loading system; voice coil motor; dynamic performance; force load simulation

---

## 1. Introduction

The concept of more electric aircraft was proposed for the replacement of centralized hydraulic power systems with distributed power-by-wire systems because of the promising benefits of higher reliability, increased dynamic performance, convenient maintainability, weight reduction and easier manufacture. The core component of the distributed power-by-wire system using electric power directly has yielded three types of approach, including the electrohydraulic actuator (EHA), the electromechanical actuator (EMA) and the integrated actuation package (IAP) [1,2]. We propose the idea of a linear-driven electrical hydraulic actuator (LEHA) which is required to be of high frequency and large output force. The linear pump is driven by a linear oscillating motor (LOM), as shown in Figure 1, so it is of great importance to have detailed research on the LOM, especially on the dynamic performance of the LOM with external force loads.

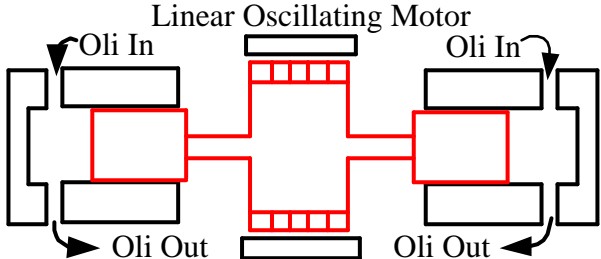

**Figure 1.** Diagram of linear pump.

The linear motor, providing reciprocating motion without any other ancillary components, was proposed in America more than a century ago. At the beginning of the exploration of the linear motor, Kemper in Germany proposed a kind of magnetically levitated vehicle propelled by linear motor. After that, LOMs related to the maglev became hot spots until the end of 1990s when the low loss short-stroke LOM became increasingly popular. Actually, linear motors have vital and typical applications on machine tool slid tables, recorders and free piston engines. Since 1999, LOMs especially LOMs have been vitally and extensively applied to air compressors, pumps, electromagnetic valves, active shock absorbers, vibrators and earphones [3–7]. The application of LOMs to the LEHA, a kind of hydraulic pump controlled actuator, leads to consequence of good reliability, high power density and convenient maintenance, which requires excellent dynamic displacement output with external load force and large force output. So it is promising to study LOM dynamic performance as well as efficiency in order to apply the LOM better to the LEHA.

Until now, most research has concentrated on force output, motor design and optimization, and control strategy of LOMs. Zhu explored the permanent magnetic array, the thrust force capability, the optimal design parameters, etc. to improve the static output force and reduce the cogging force [8–12]. As for the force control, some internal forces of the LOM such as cogging force, detent force, ripple force and force of friction have been taken consideration into the trajectory track of LOMs. LU took the cogging force into compensation control to improve the industrial gantry position control accuracy, but no external load is considered except for the friction force [13]. A detent force compensation control was conducted by Wang without any external force involved [14]. Lin had a proposal of a field-programmable gate array (FPGA)-based computed force system to improve the position control performance of the linear ultrasonic motor for various reference trajectories with no load force acting on the LOM [15]. Hwang applied the Jacobian linearization observer to the ripple force compensation control to improve the position trajectory accuracy [16]. All these research works above focused on the LOM itself, not taking multiple external force loads such as spring force, rectangular force and damping force into consideration to study the dynamic performance of the LOM. Therefore, it is meaningful and offers promising prospects to have research on the effects of multiple external force loads on the trajectory track of the LOM and it is necessary to develop a linear oscillating load system (LOLS) for the LOM.

As for force loads forms, there are four types including magnetic powder brake, compressed air, weight and voice coil motor (VCM). The magnetic powder brake, a kind of traditional and mature instrument for a rotary motor, is unsuitable for the LOM load simulation because of its ability to impose rotary damping force on the shaft. Even though a pinion and rack transmission mechanism is adapted to transforming linear motion to rotary motion, the transformed motion is in swing and affected severely, and is not ideal rotary motion, which causes unavailability of the magnetic powder brake function. Consequently, direct force loads drew researchers' attention. Compressed air is usually applied to the simulation of air load for air compressor testing [17], but this type of force load is deficient and improper for simulating the hydraulic force load because of severe elasticity and non-direction. The suspended weight can provide constant force load on the LOM. Yang added weights on the LOM as the constant force load for the study of the thrust force harmonies [18]. Cao also applied weights to a complementary and modular linear flux-switching permanent-magnet motor on the force control

research [19]. Although weight force loads have the feature of constant and stability, this type of force load would add redundant inertia to the mover of LOM, which will have serious influence on the LOM dynamic performance, and the weight force loads are limited because of the gravity limitation. The VCM was applied by Katalenic to the short stroke reluctance linear actuator for the high-precision force control [20], but the performance of the VCM was of partial development, and the output force of the VCM was small. In summary, all these research works lack dynamic performance analysis of the LOM with multiple external force loads, which limited applications and popularization of the LOM. Therefore, it is promising to design a linear oscillating loading system and search effects of different load modes and force amplitudes on the dynamic performance of the LOM, and this work will have significant influence on the application of the LOM to the LEHA, to promote the development of the More Electrical Aircraft (MEA).

In the paper, the design and prototype of the linear oscillating load system, based on the VCM, is introduced in Section 2; a mathematical model of the LOLS is given in Section 3; Simulation analysis and experimental results of the dynamic performance of the LOM are introduced in Section 4; finally, a conclusion is drawn in Section 5.

## 2. Design and Prototype of Linear Oscillating Load System

### 2.1. Voice Coil Motor (VCM)

The VCM can be constructed as two parts, identified as the mover and the stator. The mover consists of copper coils and coil frame, as well as the stator is composed of irons and magnets. According to the features of large length to ratio of the mover with the large working stroke, axially magnetized magnets are chosen as the source of the main magnet field in the stator. Irons, i.e., cover irons and back iron, are designed as the flux-conducting paths, aiming at reducing magnet resistance and flux leakages. Due to the radial flux density in the air gap, axial thrust force can be generated in the coils of electric excitation. To improve usage rate of flux and increase axial thrust force, two parts of copper coils are install in the air gap. As a conclusion, the sketch structure of the VCM can be designed as shown in Figure 2.

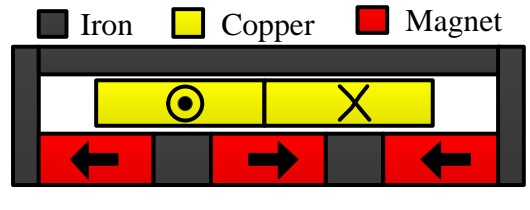

**Figure 2.** Sketch structure of the voice coil motor (VCM).

According to the sketch structure and material features of the VCM, the open-circuit flux paths are depicted in Figure 3 for detailed description. The flux produced by the end axial magnets passes through cover iron, back iron, air gap and inner iron as well as that by the middle axial magnet passes through inner iron, air gap and back iron. So the flux passes through a complete magnetic circuit, which reduces the magnet resistance as well as the flux leakage efficiently.

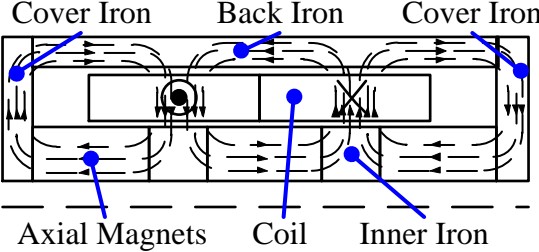

**Figure 3.** Open-circuit flux lines of the VCM.

Furthermore, mechanical structures of mover coils and magnetic array are of necessity for support and fixation. A PEEK (polyether-ether-ketone) tube with no ferromagnetism and no electrical conductivity is wound by the mover coils, which fixes the mover coils and reduces eddy current effect and mover mass effectively. A core rod of no ferromagnetism, made up of stainless steel, is installed in the inner iron rings and magnets with screw joints at the ends of the rod to fix the magnets and irons firmly. As a result, the 3D object of the VCM is draw as depicted in Figure 4.

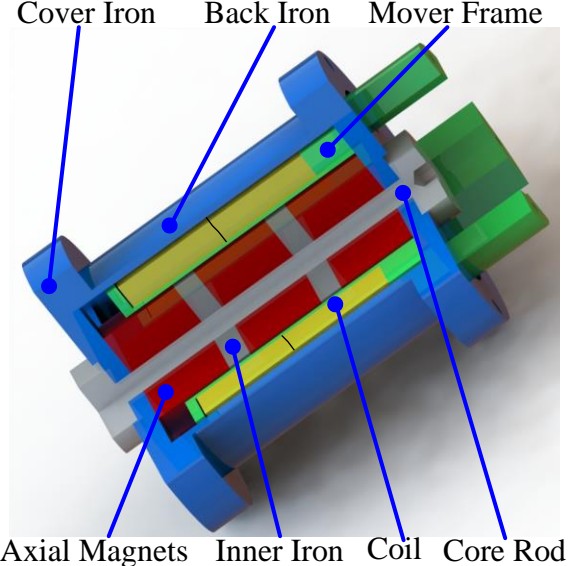

**Figure 4.** Three-dimensional (3D) structure of the VCM.

Based on the structure design above, the prototype of the linear VCM is established as shown in Figure 5. The mover consists of coils and tube frame as well the stator is composed of magnetic array, cover irons and back iron. The model number of magnets is NdFe35SH and the diameter of the coil copper wire is 1 mm.

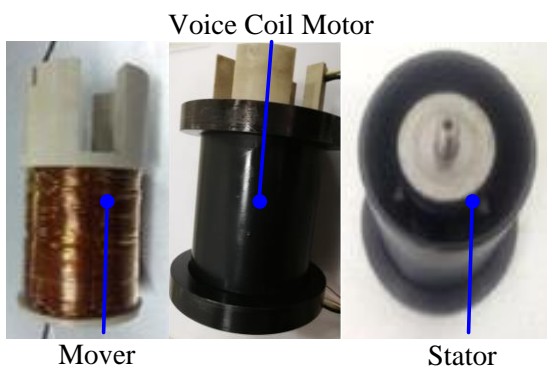

**Figure 5.** Detailed structure of the VCM.

Moreover, the comparison of the output force density between experimental results and simulation results is depicted in Figure 6, showing perfect consistency between theoretical results and experimental results and proving the validity of the proposed design. Furthermore, the fluctuation around the average force density is about 4.33%, to ensure the symmetry of load force within stroke. Meanwhile, the relationships between current and output thrust force at three locations of the mover, i.e., 0 mm, −5 mm and 5 mm, are depicted in Figure 7, demonstrating the nearly proportional relationship between thrust force and current. Because of excellent symmetry and linearity of the VCM output thrust force, the VCM can be utilized as the load motor for the LOLS, having the advantages of accurate position and force control performance.

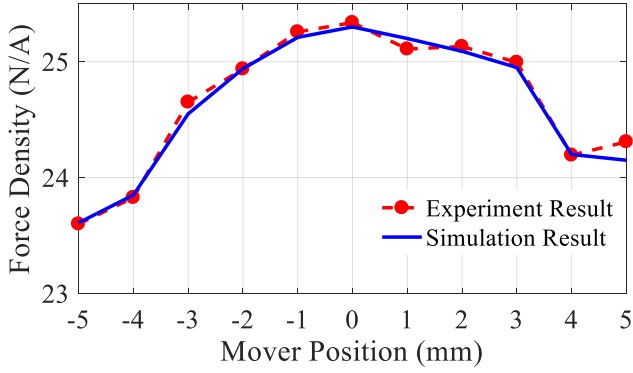

**Figure 6.** Force constant variations within stroke.

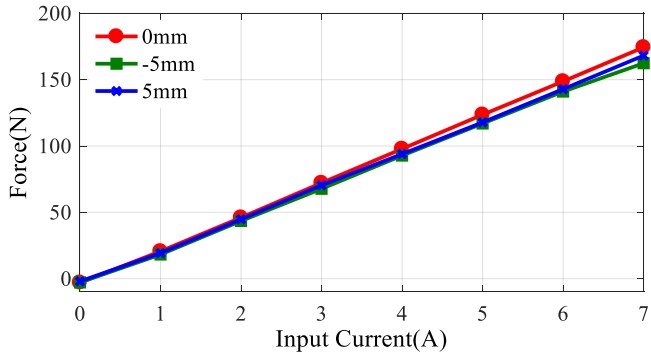

**Figure 7.** Force with different current input.

### 2.2. Linear Oscillating Load System (LOLS)

The cross-sectional view of the LOM is shown in Figure 8, and the LOM is composed of stator, mover and springs. Based on the VCM designed above, the LOLS can be designed as show in Figure 9. The LOLS is composed of three parts, i.e., VCM, sensor mechanism and LOM to be tested. All these three parts of the LOLS are fixed on an iron platform. The VCM is connected with the LOM through the sensor mechanism, so the output thrust force of the voice motor can be regarded as the simulation load. Because of the excellent symmetry and linearity of the VCM, it is possible and reliable to simulate multiple types of force loads such as spring force, damping force and rectangular force for the LOM. The position signal obtained from displacement sensor is utilized to observe velocity and acceleration, and multiple types of force load can be simulated. To identify multiple forms of force loads, the force sensor is used for detection.

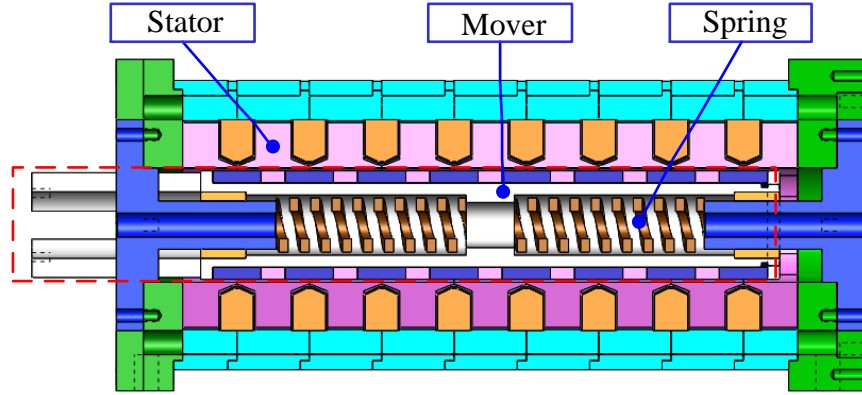

**Figure 8.** Cross-sectional view of the linear oscillating motor (LOM).

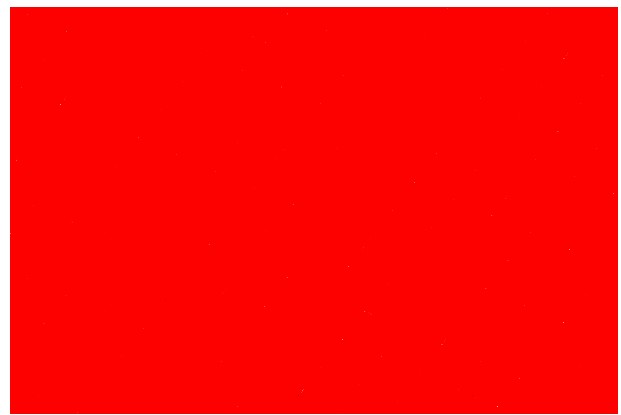

**Figure 9.** 3D structure of linear oscillating loading system.

Based on the 3D structure of the linear oscillating load system designed above, the physical linear oscillating load system is shown in Figure 10. Two aluminum connectors are utilized to link the VCM, the sensor mechanism and the LOM to be tested.

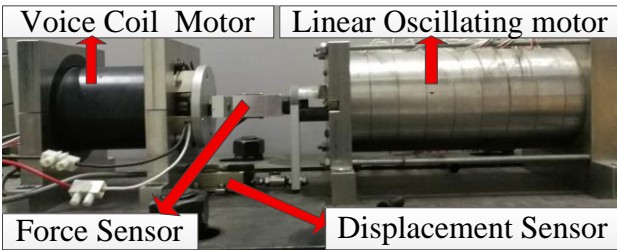

**Figure 10.** Image of linear oscillating load system.

As shown in Figure 10, movers of two linear motors and force sensor are constrained by one degree of freedom, so the mover displacement and velocity of the VCM are same to that of the LOM. Output thrust force of the VCM acts on the linear oscillating motor via the force sensor. The force types of the output thrust force can be regarded as the force load forms of the LOM to be tested, and three types of simulation force modes, i.e., spring force, damping force and rectangular force, are depicted in Figure 11.

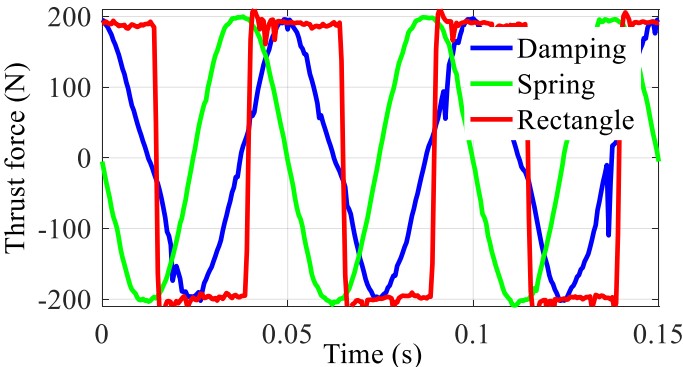

**Figure 11.** Different modes of force load.

Figure 10 is on the honor of the fact that the VCM can simulate these three types of force load forms well as the amplitude is 200 N and the frequency is 20 Hz. It is meaningful for the research of the dynamic performance analysis for the LOM with multiple external forces.

### 3. Mathematical Model of Linear Oscillating Loading System

For the LOM is of excellent symmetry and linearity, it is available to assume that the LOM is of ideal relationship between output thrust force and input current. Hence, the electromagnetic force of the LOM can be written as:

$$F_{e1} = K_{e1}i_1 \tag{1}$$

where $F_{e1}$ is the electromotive force of the LOM to be tested, $K_{e1}$ is the force constant coefficient of the LOM, $i_1$ is input current.

Analogously, the electromagnetic force function of the VCM can be written as:

$$F_{e2} = K_{e2}i_2 \tag{2}$$

where $F_{e2}$ is the output thrust force regarded as load force of the VCM, $K_{e2}$ is the force constant coefficient of the VCM, and $i_2$ is the input current.

Assume that movers are rigid bodies and ignore deformation of the force sensor, the dynamic equation of the LOLS, based on the Newton second law, can be written as:

$$F_{e1} - F_{e2} = kx + (\xi_1 + \xi_2)\mathrm{sgn}(\frac{dx}{dt}) + (m_1 + m_2)\frac{d^2x}{dt^2} \tag{3}$$

where $k$ is the spring stiffness of the LOM; $m_1$ is the mover mass of the LOM; $m_2$ is the otal mass of the VCM mover and the sensor mechanism; $\xi_1$ is the kinetic friction force of the LOM, and $\xi_2$ is the kinetic friction force of the VCM.

Similarly, the dynamic equation of the LOM can be written as:

$$F_{e1} - F_{s1} = kx + \xi_1\mathrm{sgn}(\frac{dx}{dt}) + (m_1 + m_2)\frac{d^2x}{dt^2} \tag{4}$$

where $F_{s1}$ is force magnitude between the force sensor and the LOM.

Approximately, the dynamic equation of the load mechanism can be written as:

$$F_{e2} - F_{s2} = \xi_2\mathrm{sgn}(\frac{dx}{dt}) + m_2\frac{d^2x}{dt^2} \tag{5}$$

where $F_{s1}$ and $F_{s2}$ are the coupled forces of action and reaction. Thus, the force equivalency equation, based on the Newton third law, can be written as:

$$F_{s1} = F_{s2} = F_s \tag{6}$$

As for the LOM, voltage is composed of three parts, i.e., resistance voltage, inductance voltage and electromotive force (EMF) voltage, so the voltage balance equation of the LOM can be written as:

$$U_1 = i_1 R_1 + L_1 \frac{di_1}{dt} + K_{e1} \frac{dx}{dt} \tag{7}$$

where $U_1$ is the input voltage of the LOM, $R_1$ is the total resistance accorded with series and parallel mode of the LOM coils. $L_1$ is the total inductance accorded with $R_1$.

Similarly, the voltage balance equation of the VCM can be written as:

$$U_2 = i_2 R_2 + L_2 \frac{di_2}{dt} + K_{e2} \frac{dx}{dt} \tag{8}$$

where $U_2$ is the input voltage of the VCM, $R_2$ is the total resistance accorded with parallel mode of the VCM coils. $L_2$ is the total inductance accorded with $R_2$.

In this paper, three typical types of force loads are set to be achieved, i.e., spring force, damping force and rectangular force. For $F_{e2}$ is regarded as the load force, the relationship between $x$ and $i_2$ should be given as follows:

$$i_2 = f(x, \frac{dx}{dt}) \tag{9}$$

If simulation force is spring force, the relationship between $x$ and $i_2$ can be written as:

$$f(x, \frac{dx}{dt}) = K_s \cdot x \tag{10}$$

$K_s$ is the spring stiffness of the spring simulation force. If the simulation force is a damping force, the relationship between $x$ and $i_2$ can be written as:

$$f(x, \frac{dx}{dt}) = \kappa \cdot \frac{dx}{dt} \tag{11}$$

$\kappa$ is damping coefficient of the damping simulation force.

If the simulation force is a rectangular force, the relationship between $x$ and $i_2$ can be written as:

$$f(x, \frac{dx}{dt}) = K_r \cdot \text{sgn}(\frac{dx}{dt}) \tag{12}$$

$K_r$ is the gain coefficient of the rectangular simulation force.

## 4. Simulation and Experiment Analysis

According to the mathematical model built above, the block diagram of the LOLS can be constructed, as shown in Figure 12.

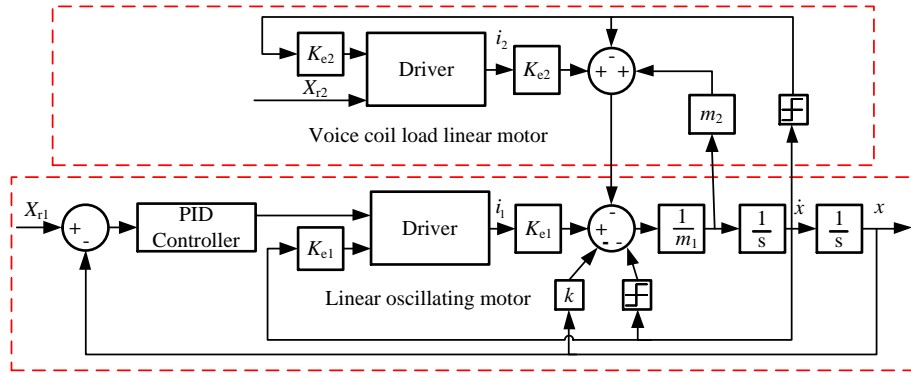

**Figure 12.** Block diagram of the linear oscillating load system (LOLS).

As can be seen in Figure 11, the block diagram can be separated as two parts, i.e., the VCM to provide the thrust load force at the upper block diagram, and the LOM to be tested at the lower block diagram. $X_{r1}$ is the reference signal of the LOM and $X_{r2}$ is the input signal of the VCM, accorded with f(x), so three types of load forces, i.e., spring force, damping force and rectangular force, can be simulated. Based on the block diagram of the LOLS designed above, both simulation analyses and experimental verification can be carried out to research the dynamic performance of the LOM with a multiple external force load.

*4.1. Simulation Analysis*

Main parameters values of the LOLS are shown in Table 1.

**Table 1.** Main parameters of the LOLS.

| Simulation Model Parameters | Value |
|---|---|
| Force constant coefficient of the LOM $K_{e1}$/(N/A) | 32 |
| Force constant of the VCM $K_{e2}$/(N/A) | 25 |
| Spring stiffness k/(N/m) | 18,141 |
| Damping force of the LOM $f_{c1}$/N | 50 |
| Damping force of the VCM $f_{c2}$/N | 10 |
| Mover mass of the LOM $m_1$/kg | 1.28 |
| Mover mass of loading mechanism $m_2$/kg | 1.1 |
| Resistance of the LOM $R_1$/$\Omega$ | 2.6 |
| Resistance of the VCM $R_2$/$\Omega$ | 1.2 |
| Coil inductance of the LOM $L_1$/H | 0.024 |
| Coil inductance of the VCM $L_2$/H | 0.0033 |

Based on the mathematical model and the block diagram of the LOLS built above, the simulation analysis of the dynamic performance of the LOM with external spring force load can be finished as shown in Figure 13, and the simulated spring stiffness is 133.33 N/mm, 66.67 N/mm and 33.33 N/mm corresponding to amplitude force of 400 N, 200 N and 100 N, respectively.

As can be seen in Figure 13, spring forces affect both phase responses and amplitude responses as the phases advance by 8.78°, 5.62° and 0.65°, and the amplitude responses reduce by 5.5%, 4.1% and 2.7%, corresponding with amplitude forces of 400 N, 200 N and 100 N respectively, compared with no load. So phase advances and amplitude attenuation extend as the simulated spring stiffness becomes bigger.

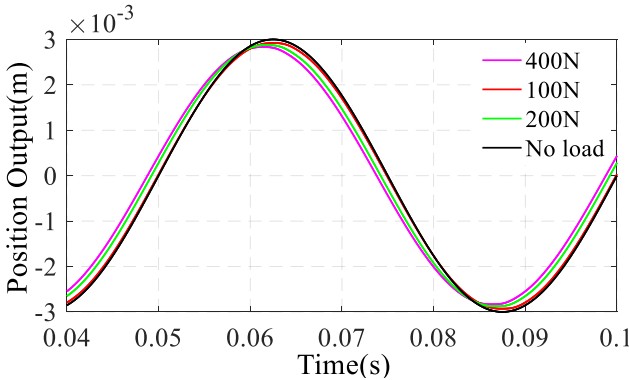

**Figure 13.** Position outputs under different spring loads.

Similarly, the dynamic performance of the LOM with external damping force loads is depicted in Figure 14, corresponding with the amplitude damping forces of 200 N and 400 N respectively.

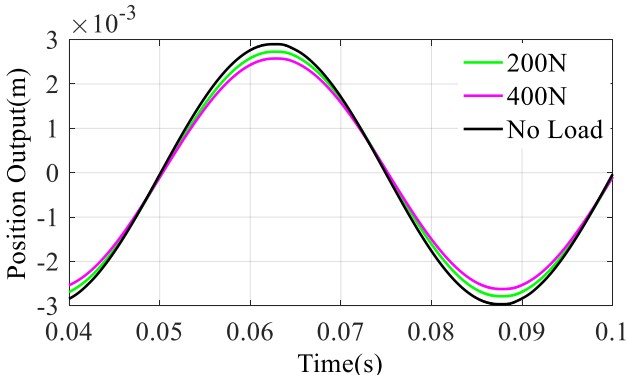

**Figure 14.** Position outputs under different damping loads.

As it can be seen in Figure 14, damping forces have an effect on amplitude responses particularly as the amplitude responses reduce by 6.1% and 11.5% corresponded with the amplitude damping forces 200 N and 400 N respectively, compared with no load. Damping forces merely affect the amplitude responses; as the amplitude damping force is bigger, the amplitude attenuation is bigger.

Similarly, the dynamic performance of the LOM with external rectangular force loads is depicted in Figure 15, corresponding to the amplitude rectangular forces of 200 N and 400 N respectively.

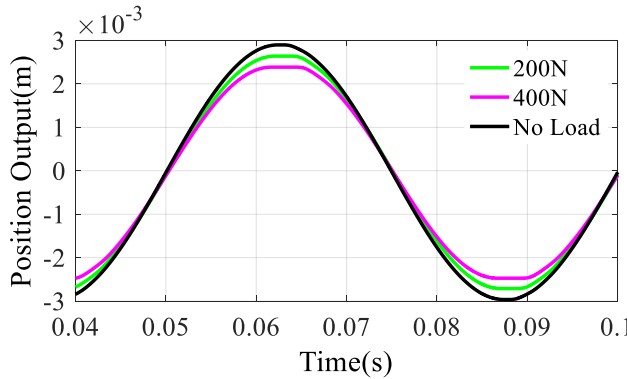

**Figure 15.** Position outputs under different rectangular loads.

As can be seen in Figure 15, rectangular forces have effects on the position outputs, especially on the amplitude responses as these reduce by 16.6% and 8.5% corresponding with the amplitude

rectangular forces of 200 N and 400 N respectively, compared with no load. Rectangular forces have effects on the amplitude responses more seriously than effects of damping forces at the same force load amplitude.

For these three types of force load, damping force and rectangular force only has effects on the amplitude response as the bigger the amplitude force is, the bigger the amplitude attenuation, while the spring force have effects not only on the amplitude response but also on the phase response as the bigger the amplitude force is, the bigger the amplitude attenuation and phase advance are. Because the spring stiffness is increased owing to the added spring load, to raise the cut-off frequency of the LOM leads to smaller phase lag.

### 4.2. Experiment Results

Based on the linear oscillating load system built above, experiments are finished to analyze the dynamic performance of the LOM with multiple external force loads, i.e., spring load, damping load and rectangular load. Position outputs of the LOM under multiple force loads are depicted in Figures 16–18, respectively.

As can be seen in Figure 16, the experimental result indicates that the spring force load affects both phase responses and amplitude responses. As the phases advance by 7.2°, 14.4° and 25.2°, and the amplitude attenuations increase by 7.4%, 14.9% and 19.2%, according to different amplitude spring force loads of 100 N, 200 N and 400 N, respectively. Experimental results and simulation results shown in Tables 2 and 3 have the same effects on the amplitude and the phase lag to be smaller. Although impacts between simulation and experiment are different because of different closed loop controls and differences between physical model and mathematical model, the trend is similar.

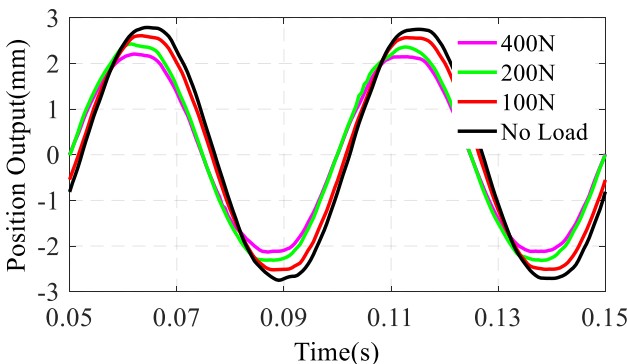

**Figure 16.** Experimental position outputs under spring forces.

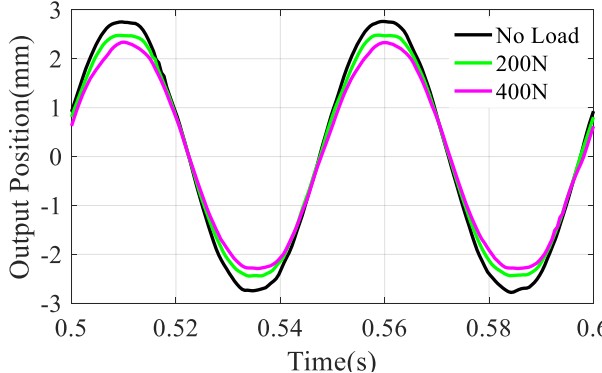

**Figure 17.** Experimental position outputs under damping forces.

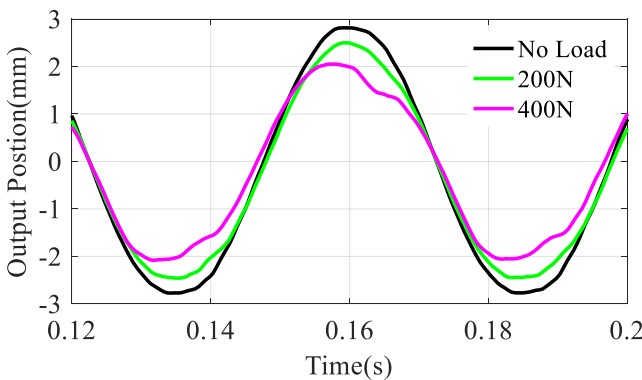

**Figure 18.** Experimental position outputs under rectangular forces.

**Table 2.** Amplitude attenuation with spring force load.

| Amplitude | Simulation | Experiment |
|---|---|---|
| 100 N | 2.7% | 7.4% |
| 200 N | 4.1% | 14.9% |
| 400 N | 5.5% | 19.2% |

**Table 3.** Phase advance with spring force load.

| Amplitude | Simulation | Experiment |
|---|---|---|
| 100 N | 0.65° | 7.2° |
| 200 N | 5.62° | 14.4° |
| 400 N | 8.78° | 25.2° |

As can be seen in Figure 17, damping forces mainly compress the amplitude of the output position as the amplitude attenuations increase by 9.3% and 16.3% accorded with the amplitude damping force loads of 200 N and 400 N, respectively.

As can be seen in Figure 18, rectangular forces mainly compress the amplitude of the output position as the amplitude attenuations increase by 9.8% and 26% according to the amplitude rectangular force loads of 200 N and 400 N, respectively. Experimental results and simulation results of damping force and rectangular force loads shown in Tables 4 and 5 indicate that damping force and rectangular force mainly have effects on the amplitude attenuation.

**Table 4.** Amplitude attenuation with damping force load.

| Amplitude | Simulation | Experiment |
|---|---|---|
| 200 N | 6.1% | 14.9% |
| 400 N | 11.5% | 19.2% |

**Table 5.** Amplitude attenuation with rectangular force load.

| Amplitude | Simulation | Experiment |
|---|---|---|
| 200 N | 8.5% | 9.8% |
| 400 N | 16.5% | 26% |

The analyses above are to show dynamic performance of the LOM with multiple types of force loads, but the robustness of the controller used in the closed loop of the LOM is inadequate. If an external force compensation control strategy is applied to the closed loop control of the LOM, the amplitude of the rectangular force can be 800 N, as depicted in Figure 19.

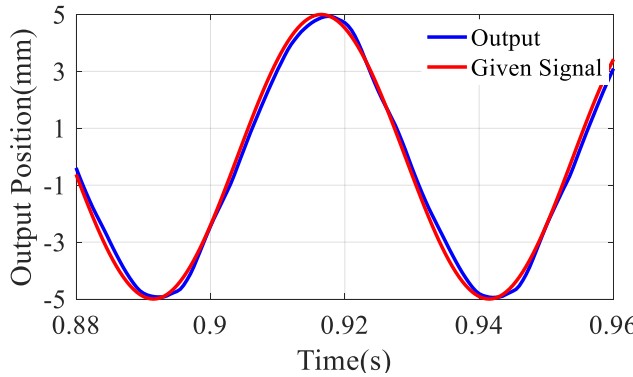

**Figure 19.** Output position with 800 N amplitude of rectangular force.

## 5. Conclusions

At present, one of the hindrances for the application development of the LOM is the lack of load testing system to have in depth research on the dynamic performance of the LOM with enough external force loads, in order to exhibit the ability of force load tolerance to achieve accurate output position of the LOM. In this paper, a linear oscillating load system is proposed and designed to analyze the dynamic performance of the LOM with external load forces at short stroke and high-frequency working condition. Both physical and mathematical models are built, and both experiment and simulation results indicate the same effects on the dynamic performance of the LOM as the spring loads influence both amplitude responses and phase responses to lead more amplitude attenuation and less phase lag, and damping loads and rectangular loads mainly reflect more amplitude attenuation. Furthermore, when proper external force compensation control is applied, the tolerance force can be of 800 N amplitude rectangular force load, indicating that the LOM has the ability to maintain an accurate output position with large force load at the resonant frequency of the LOM, which provides the basis for research into the linear driven electrical hydraulic actuator.

**Author Contributions:** Conceptualization: Z.J.; Data curation: Y.C.; Investigation: L.Y., X.L. and L.Z.; Methodology: Y.C. and X.L.; Supervision: Z.J.; Visualization: Y.C.

**Funding:** This work was supported by the National Natural Science Foundation of China under grant 51890822, 51875013 and 51575026, National Key Basic Research Program of China under grant 2014CB046400, the National Key R&D Program of China under grant 2017YFB1300400, the Fundamental Research Funds for the Central Universities, Science and Technology on Aircraft Control Laboratory, and Ningbo Institute of Technology, Beihang University.

**Conflicts of Interest:** The founding sponsors had no role in the design of the study; in the collection, analyses, or interpretation of data; in the writing of the manuscript, and in the decision to publish the results.

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
