# Peer review of "Design and Analysis of Novel Linear Oscillating Loading System"

_applsci, doi:10.3390/app9183771_

Round 1
Reviewer 1 Report
This paper presents a design and analysis of novel linear oscillating loading system. The linear oscillating loading system platform is built and experiments are proceeded to verify the simulation analyses about the dynamic performance and efficiency with respect to different external force loads. The content of this paper is interesting and well supported with analysis evidence. Nevertheless, this paper needs some revision. The paper needs the following changes before it could be accepted for publication.
page 1 : In Figure 1, the structure of the LOM needs to be expressed in more detail. Page 3 (Line 96~) : In Section 2.1, a specific spec. of the VCM to be used as a load is required. This appears to be effective when expressed in tables. Page 5 : In Figure 6, the change in force density occurs up to 4.33% depending on the mover position. In this study, a criterion for the change of the force density required for the load is required and the basis for this should be given. Page 5 : In Section 2.2, a specific spec. of the LOM or LOLS in table form is required. Page 5 : In Section 2.2, a specific cross-sectional view of the LOM is needed. Figure 8 can be replaced with Figure 9. Page 10 : It is required to use table for quantitative comparison of various analytical values and experimental values. In addition, the cause analysis of the difference between the analytical value and the experimental value needs to be clarified more clearly.
Reviewer 2 Report
line 17 "march" or "match"?
Fig.1 Oli means what? It's Oil?
line 39 "by" or "in"?
line 92 "SectionIII" or "Section III"?
line 130 experimental results should be detailed...
Fig.8 what kind of positio sensoe and force sensor were used? Give more information
lines 186 and 187 refering to equation 3 variables "damping coefficient of the LOM" and "kinetic friction force of the LOM" share the same greek letter?
line 201 refering to equation 7 definition of variable Ke1 is missing
line 205 Ke1 or Ke2 "is EMF coefficient of the VCM"?
line 233 in Tab.1 parameters were obtained from?
line 295 what control strategy was applied? Give more information
Author Response
Dear Professors,
The authors would like to sincerely appreciate your constructive comments on our paper, “Design and analysis of novel linear oscillating loading system” (Manuscript ID.: applsci-568322). These comments help us to have a broader view of research issues as well as to further consolidate the paper.
Following the valuable suggestions, we have carefully checked through the manuscript for a few rounds, and made quite a lot of revisions accordingly. The major revisions are highlighted in the manuscript. Enclosed are our replies to your comments other than spelling mistakes and typographical errors.
Sincerely appreciate your kind assistance. Thank you very much!
Replies are below.
1.Mistakes have been corrected.
2.Position sensor is a potentiometer and force sensor is an S-shape sensor.
3.We use force compensation control to make trajectory performance fitted at large force load.
